# *EA3D*: Online Open-World 3D Object Extraction from Streaming Videos

**Xiaoyu Zhou**[1†]  **Jingqi Wang**[1†]  **Yuang Jia**[1]  **Yongtao Wang**[1*]
**Deqing Sun**[2]  **Ming-Hsuan Yang**[2,3]

[1]Wangxuan Institute of Computer Technology, Peking University
[2]Google DeepMind  [3]University of California, Merced

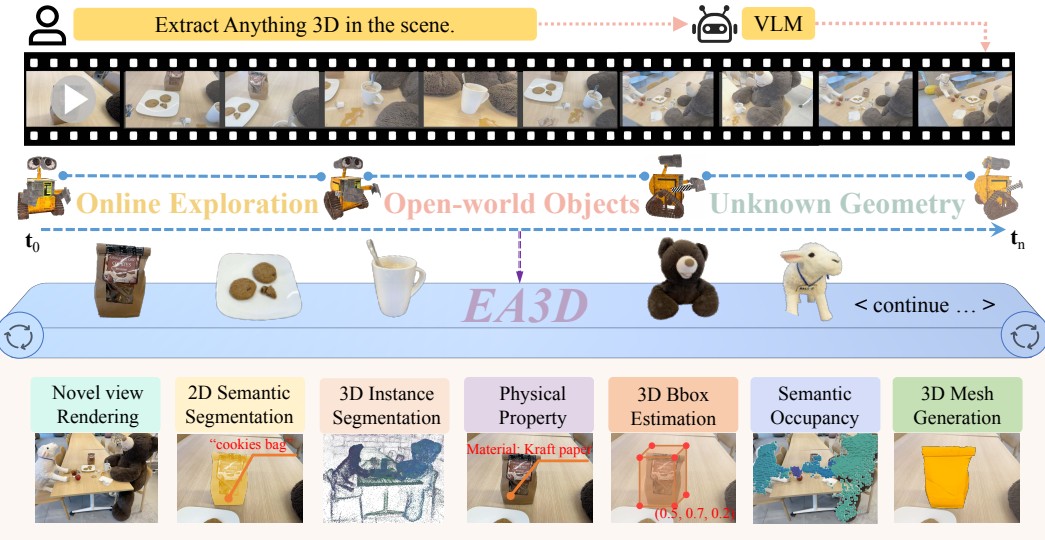

Figure 1: Illustration of *EA3D*, which enables online open-world 3D object extraction. Given a streaming video as input with unknown geometry, pose, or semantics, EA3D performs online and simultaneous scene interpretation and geometry reconstruction, enabling multi-task understanding and modeling of any 3D objects in the scene.

## Abstract

Current 3D scene understanding methods are limited by offline-collected multi-view data or pre-constructed 3D geometry. In this paper, we present *ExtractAnything3D* (EA3D), a unified online framework for open-world 3D object extraction that enables simultaneous geometric reconstruction and holistic scene understanding. Given a streaming video, EA3D dynamically interprets each frame using vision-language and 2D vision foundation encoders to extract object-level knowledge. This knowledge is integrated and embedded into a Gaussian feature map via a feed-forward online update strategy. We then iteratively estimate visual odometry from historical frames and incrementally update online Gaussian features with new observations. A recurrent joint optimization module directs the model's attention to regions of interest, simultaneously enhancing both geometric reconstruction and semantic understanding. Extensive experiments across diverse benchmarks

---

[*]Corresponding author.

39th Conference on Neural Information Processing Systems (NeurIPS 2025).

and tasks, including photo-realistic rendering, semantic and instance segmentation, 3D bounding box and semantic occupancy estimation, and 3D mesh generation, demonstrate the effectiveness of EA3D. Our method establishes a unified and efficient framework for joint online 3D reconstruction and holistic scene understanding, enabling a broad range of downstream tasks. The project webpage is available at https://github.com/VDIGPKU/EA3D.

# 1   Introduction

To see is, as famously defined by David Marr [25], "*to know what is where by looking.*" For an autonomous agent, such as a robot, operating in an unfamiliar environment, this translates into formidable challenges. Imagine a robot entering a new room, observing and understanding its surroundings on the fly (Fig. 1). It faces an unknown quantity and variety of objects (**open world**) and needs to process unfamiliar 3D geometry (**unknown geometry**) in a streaming mode (**online exploration**). To effectively navigate and interact within such a dynamic 3D space, the robot must be able to dynamically construct open-world 3D representations of the scene. Concurrently, it must comprehend the geometric structures and physical properties of the objects it encounters and perceptively model the motion states of all semantic entities within complex, evolving environments.

While Vision-Language Models (VLMs) [13, 57, 21] show impressive results on 2D open-world understanding , they struggle in 3D domains, exhibiting view inconsistencies[58, 11], geometric misalignment[1], and inability to handle occlusions. A straightforward solution is to lift 2D VLM outputs into 3D using scene geometry [49, 56, 14], but this requires pre-constructed 3D geometry, annotated datasets for training, and still suffers from 3D-2D misalignment issues. Recent differentiable rendering frameworks like NeRF [27, 36] and 3DGS [15, 54, 8] enable joint 3D scene understanding by optimizing 3D representations with pixel-level pseudo-labels[16, 52, 65, 31]. However, these offline approaches require complete multi-view images and time-consuming multi-stage processes.

In this paper, we introduce ***ExtractAnything3D*** (EA3D), an online open-world scene understanding framework that simultaneously explores, reconstructs, and interprets the 3D geometry and semantic knowledge of a scene. Similarly to human perception, our system starts processing streaming visual inputs as soon as it enters a room, reconstructing and understanding the current scene online based on historical observations and prior knowledge. As new frames emerge, they progressively reveal more comprehensive spatial information, enriching the internal knowledge base and allowing the system to infer occluded regions via novel view synthesis.

Specifically, we utilize VLMs to openly interpret object categories and physical properties from the emerging frame while dynamically maintaining a semantic cache. We then combine features from multiple visual foundation models with semantic cues to construct a dynamically updated knowledge-integrated feature map. The knowledge-integrated features are embedded into Gaussian representations through a fast feedforward step and are updated jointly over time. To incrementally extract both geometry and knowledge of 3D objects in an online manner, we construct Online Feature Gaussians, consisting of two core components: online visual odometry and online Gaussian updating. Benefiting from a recurrent joint optimization strategy, our proposed Online Feature Gaussians dynamically extract any 3D objects in the scene, facilitating multiple tasks including photo-realistic rendering, semantic and instance segmentation, physical property analysis, and geometric reasoning (e.g., 3D bounding boxes, semantic occupancy, and 3D mesh generation). EA3D thus establishes a unified and efficient framework for joint online 3D reconstruction and holistic scene understanding, enabling a wide range of downstream tasks.

The contributions of this work are: 1) We propose a unified online open-world 3D objects extraction framework enabling simultaneous online reconstruction and understanding without geometric or pose priors. 2) Taking streaming video as input, our method effectively leverages historical knowledge to guide 3D object extraction at the current observation, enabling online joint updates of integrated features and delivering high-quality, efficient geometric reconstruction and scene understanding. 3) Our method supports a broad set of tasks, including photo-realistic reconstruction and rendering, semantic and instance segmentation, 3D bounding box construction, semantic occupancy estimation, and 3D mesh generation, consistently achieving good performance across multiple benchmarks.

## 2   Related Work

**Open-World Foundation Model.** When exploring the real world, the quantity and categories of 3D objects remain unknown in unbounded environments. Recent advances in Vision-Language Models (VLMs) and Vision Foundation Models (VFMs) have significantly advanced open-world interpretation of 2D images. VLMs [13, 21, 57, 42] effectively fuse visual and textual cues for Visual Question Answering (VQA), while SAM-based [17, 33] and CLIP-based methods [9, 22, 51] excel in generalized semantic segmentation and instance detection. However, these methods suffer from severe multi-view inconsistencies and semantic ambiguities, especially for small objects, due to their limited geometric awareness. They also struggle with spatial occlusions and suffer from memory degradation over time. To overcome these challenges, we propose an online, synchronized framework for joint reconstruction and understanding, where 2D foundational features are implicitly aligned throughout the online reconstruction process. Our framework leverages online embedding from VFMs and recurrent joint optimization to seamlessly align 2D knowledge with 3D geometry, ensuring coherent consistency across the 3D domain.

**3D Scene Understanding.** Current 3D scene understanding methods broadly categorized into two groups: (1) methods that operate on known 3D geometry—such as point clouds, depth maps, or meshes; and (2) methods that infer scene semantics while reconstructing the 3D geometry. Methods like [30, 40] and [55, 3] extract semantics via 2D-to-3D lifting, but all depend on pre-built 3D geometry and costly semantic annotations. Recent approaches address this limitation by jointly reconstructing and segmenting 3D scenes through differentiable rendering. NeRF [16, 2] and 3DGS-based methods [52, 65, 31, 18] leverage pseudo-labels to jointly optimize appearance and semantics via 2D supervision. However, both types of methods are inherently offline, relying on full scene observations before reconstruction and interpretation. In real-world settings, agents dynamically explore and progressively understand scenes. To address this gap, we propose an online framework for simultaneous scene reconstruction and understanding. Our method efficiently builds 3D objects while delivering high-quality semantic interpretation. Guided by evolving 3D geometry, it enables comprehensive extraction of open-world objects.

**Online Reconstruction.** Recent advances in 3DGS [15, 54] have demonstrated remarkable capabilities in photo-realistic rendering and have been extended to a range of downstream applications, including robotic manipulation [59, 24, 37], dynamic scene reconstruction [43, 63, 12, 50], and 3D content generation [5, 64, 34]. However, vanilla 3DGS requires prolonged optimization and offline training with access to full video sequences, limiting its practicality in real-world scenarios.

To address these limitations, recent methods [39, 10, 48, 20] have proposed streaming extensions of 3DGS that significantly reduce training time and memory consumption. However, they rely on multi-view videos and pre-computed global poses, which are often impractical in real-world settings. SLAM-based approaches [26, 19] also enable online scene reconstruction but rely on sparse keyframe tracking and expensive post-refinement, limiting their ability to capture fine-grained geometry and semantics. In a related effort, an online Gaussian-based method [45] has been proposed for scene occupancy prediction. However, it is tailored for a specific task, fails to achieve photo-realistic rendering, and suffers from prohibitively expensive training costs. To overcome these challenges, we propose a novel online Gaussian optimization strategy based on knowledge feature guidance, enabling joint reconstruction and understanding of scenes in an on-the-fly manner.

## 3   Method

As shown in Fig. 2, the proposed ExtractAnything3D (EA3D) enables open-world 3D object extraction through three key components: **(a)** Knowledge extraction and integration, leveraging VLMs and multi-level VFMs for open-world understanding, integrating knowledge feature maps with an online cache and dynamically embed them into Gaussians via a feedforward way (Sec 3.1). **(b)** Online visual odometry for fast pose estimation and geometric initialization, along with online feature Gaussians that incrementally reconstruct object geometry and transfer knowledge online (Sec 3.2). **(c)** Joint optimization that continuously updates 3D object representations by fusing current observations with historical features (Sec 3.3). EA3D supports a wide range of 3D tasks.

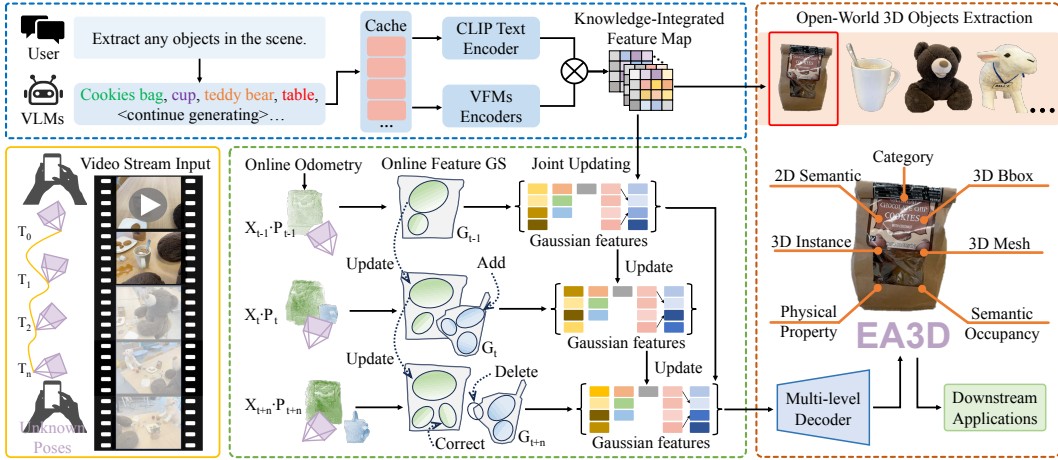

Figure 2: **Framework of EA3D.** Given a streaming video without poses or labels, EA3D first leverages VLMs to identify all potential objects and their physical attributes, while maintaining a dynamic semantic cache to track newly emerging categories. We then use multi-level VFMs to extract knowledge-integrated feature maps from each frame and embed them into Gaussian primitives via a feedforward way. We perform online visual odometry estimation, and incrementally reconstruct geometry and infer knowledge through our online feature Gaussians. A recurrent joint optimization fuses current observations with historical features to continuously update the Gaussians. EA3D supports a wide range of 3D perception tasks and shows strong potential for downstream applications.

## 3.1 Knowledge-Integrated Feature Map

Given a streaming video, we first extract object-level knowledge by dynamically interpreting the scene frame by frame using 2D vision foundation models (VFMs). However, current 2D foundational vision models lack geometric awareness of 3D scenes, leading to significant multi-view inconsistencies and ambiguities, especially in occluded regions. To tackle this challenge, we propose implicitly aligning foundational visual features in 3D space through a multi-view reconstruction pipeline based on Gaussian Splatting (GS). Each 3D representation primitive is embedded within a knowledge-integrated feature map, utilizing a feed-forward online update strategy.

**Open-world interpretation by VLMs.** VLMs [13, 57, 42] have shown exceptional open-world understanding in 2D images. Given an image $I$ observed at timestep $t$, we first use VLMs to identify all instances and their semantics within the image. In an open-world scene, the number and categories of objects are unknown. We use the prompt "Find and list all the possible objects in the given image" to capture any potential objects. Considering the continuously evolving number and semantics of objects in a streaming video, we dynamically maintain an online semantic cache $\Omega$. The online semantic cache takes input of class prompts from VLMs of the current frame, updates the semantics of newly emerged objects, and embeds them into a continuous vector $T \in \mathbb{R}^{1 \times V}$ using a pretrained text encoder from CLIP [60, 53], where $V$ denotes the changeable dimension of the vector space.

**Semantic feature map.** Despite VLMs providing comprehensive open-world interpretation, they exhibit poor visual localization ability. To address this, we leverage foundational vision models [9, 29, 33] to obtain pixel-level segmentation masks and visual features. Given a newly observed image and the online semantic cache, we utilize a pretrained CLIP visual encoder [9] and the Grounded-SAM encoder [35] to generate pixel-wise latent visual feature representations corresponding to each semantic. However, these features contain non-negligible noise and redundant information, which interfere with instance-level segmentation. Therefore, we compute the similarity of each category with semantic features using the embedded continuous vector, generating a binary mask for each category. This mask is then used to aggregate the extracted features using k-nearest neighbors. We then normalize and integrate the semantic features $\mathbf{S} = T \times \mathbf{f}_{sem}$ from different encoders, $\mathbf{f}_{sem}$ denotes the embedded semantic features, and update them into the online semantic cache.

**Physical Property.** Based on the online semantic cache and 2D priors from VLMs, we also enable the analysis of objects' physical properties. Inspired by [38, 7], we extend the text prompts to extract object-level and part-level physical properties from VLMs, corresponding to the previously obtained

semantics. We then encode the physical attribute features as a variable-length vector $\mathbf{Y}$ with a learnable prompt $y_1, \ldots, y_n$, and fuse it into the online semantic cache.

**Feature map embedding.** Vanilla Gaussian Splatting [15] represents the geometry through a collection of GS parameters, including position $\mu$, covariance matrix $\Sigma$, opacity $o$, and spherical harmonics coefficients to represent appearance. To synchronize the constructing and understanding of the 3D objects, we add an additional knowledge-integrated feature to each Gaussian. Our method integrates VLM priors, foundational visual features, and inter-track cues, combining the strengths of both appearance and geometry. Specifically, we employ a fast feedforward step to embed the knowledge features encoded by visual foundational models into the Gaussian representations. Retrieved from the online semantic cache and dynamically updated, these knowledge features exchange information across streaming frames over time. Given an emerging video frame $I_t$ at time $t$, the integrated knowledge feature map $\mathbf{F}_t^{map}$ can be formulated as:

$$\mathbf{F}_t = \sum_{i \in N, j \in N} \mathbf{X_{i,j}^{self}} \cdot \mathbf{S_{i,j}}(\mathbf{T_k}\,;\mathbf{Y_{i,j}}) \cdot \mathbf{C_t}, \tag{1}$$

where $\mathbf{F}_t$ is the integrated feature map of current frame $I_t$, $\mathbf{S_{i,j}}$ denotes the semantic features and $\mathbf{T_k}\,;\mathbf{Y_n}$ are semantic category and physical property tags. $i, j$ denote the pixel coordinates, $\mathbf{X_{i,j}^{self}}$ and $\mathbf{C_t}$ represent the corresponding point map and confidence map, as introduced in 3.2. Inspired by [46], we then compute the matching distributions of two consecutive video frames:

$$\mathbf{M}_{t,t-1} = \text{Softmax}(\frac{\mathbf{F_t}\mathbf{F_{t-1}}^{\mathsf{T}}}{\|\mathbf{F_t}\|\|\mathbf{F_{t-1}}^{\mathsf{T}}\|}), \tag{2}$$

where $\mathbf{F_t}, \mathbf{F_{t-1}} \in \mathbb{R}^{H \times W \times D}$ are the feature maps of two adjacent keyframes, where $H$, $W$ and $D$ denote height, width and feature dimension, respectively. $\mathbf{M}_{t,t-1} \in \mathbb{R}^{H \times W \times H \times W}$ is the matching distribution between two adjacent keyframes. Based on the guidance from the matching distributions, we continuously propagate the Gaussian features from the previous view to the current frame via a single forward warping, along with their corresponding knowledge feature maps. This ensures the continuity of knowledge transfer through a simple yet effective forward Gaussian transformation. We further provide a detailed comparison of our knowledge-integrated feature embedding against existing feature Gaussian methods [32, 61, 62] in the Appendix.

**Multi-level decoder for downstream tasks.** Benefiting from the knowledge-integrated feature map, the Gaussian features achieve a unified representation of object geometry and semantics. We then employ a multi-level decoder to decode the Gaussian primitives into diverse outputs, including appearance (i.e., RGB), semantics, physical properties, 3D position, depth map, 3D bounding boxes, and semantic occupancy.

## 3.2 Online 3D Objects Extraction

Suppose we are walking into a room—the construction and understanding of the 3D space begin the moment we step inside and continuously evolve as we explore. To enable this capability, we propose online feature Gaussians, which support incremental extraction of both geometry and knowledge of 3D objects in an online manner. This framework comprises two core components: 1) **Online visual odometry**, which iteratively generates and updates the poses as new frames are observed; 2) **Online Gaussian updating**, which leverages past observations to rapidly reconstruct and understand the current scene, while dynamically correcting previous misconceptions based on new observation.

**Online Visual Odometry.** Given an RGB video stream $\{I_t\}_{t=0}^N$ without camera pose, we first incrementally estimate the camera pose of the current frame based on a regression of the keypoint graph $(\mathcal{V}, \mathcal{E})$. Each graph node $\mathcal{V}_t$ corresponds to the frame $I_t$ at timestep $t$, and contains the 6-DoF pose $P_t$, pointmap $X_t$, and inverse depth $D_t$. The graph edges $\mathcal{E}$ denotes the correlation between the current frame and historical frames, with corresponding confidence maps $C_t$. We use Cut3R [41], a learning-based odometry method, in combination with [23] to estimate the initial pointmap and confidence map. Unlike concurrent work [26, 19], we integrate the dense pixel-level point map generated by Cut3R with sparse points from [23] to more effectively capture the tiny objects in the scene. However, the poses estimated by Cut3R introduce noticeable biases and errors, which accumulate over time. Therefore, we maintain an online keypoint graph and iteratively update it during reconstruction as new frames are processed. Inspired by the local bundle adjustment

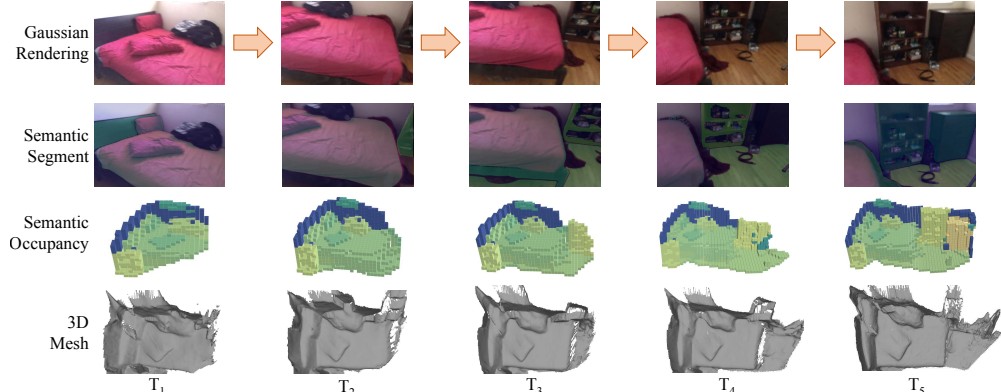

Figure 3: **Visualization of online Gaussian on Scannet [6].** EA3D processes streaming video to incrementally reconstruct while understanding. Historical features guide fast reasoning of current semantics and geometry, while new observations recurrently refine ambiguities and occlusions.

optimization [28] problem, we use a cost function adopted from [4] over the keypoint graph to minimize the reprojection error and update poses for the current frame.

**Online Gaussian Updating.** Streaming video enables dynamic observation of 3D objects through continuously emerging views, allowing previously under-observed regions to be completed and occlusion-induced ambiguities to be resolved. Inspired by this, we incrementally add feature Gaussians per frame to refine existing geometry and extract new objects. Our approach builds upon HiCoM [10], a streaming GS method designed for multi-view video reconstruction, but overcomes its reliance on predefined poses and multi-view inputs, making it suitable for fully online settings while addressing geometric and semantic challenges.

To overcome these limitations, we develop a semantics-aware online Gaussian update strategy that incrementally adds and adjusts Gaussians based on historical memory and current observations. We initialize Gaussians at timesteps 0 and 1. For each new frame, we back-project the online-estimated inverse depth map $D_t$ and pointmap $X_t$ into 3D to obtain an initial point cloud $\Phi$ for object $O \in \Omega$, which is used to initialize the corresponding Gaussians. To reduce redundancy, we adopt the transition strategy from [10, 39], assigning each Gaussian a shared translation vector and rotation quaternion within co-visible regions to maintain inter-frame consistency. For newly observed areas, we introduce new Gaussians with means $\mu_i$ initialized from the point cloud, while other attributes are optimized directly. Due to changes in occlusion, some high-opacity ellipsoids may emerge that no longer contribute to specific 3D objects, and we remove them accordingly. Additionally, we apply a one-step splitting strategy to enable adaptive Gaussian growth based on gradients, improving the representation of under-reconstructed regions. Gradients from the entire scene are finally backpropagated to jointly optimize both Gaussian parameters, features and camera poses.

### 3.3 Recurrent Joint Optimization

During online 3D object extraction, geometric reconstruction and scene understanding mutually reinforce each other. Scene knowledge priors guide the model to focus on areas of interest, while detailed geometry aids in correcting spatial inconsistencies in the priors. Notably, our method enables online joint optimization, without the need for additional post-refinement [26, 19].

**Semantic-aware adaptive Gaussian.** To leverage the correlation between object semantics and geometry, we design an adaptive semantic-awareness regularization to guide Gaussian scale adjustment:

$$\mathcal{L}_\delta = \sum |\delta_i - \bar{\delta}| F_{sem}^q, \tag{3}$$

where $\delta_i$ is the scale of the $i$-th Gaussian, and $\bar{\delta}$ is the mean scale of the particular semantic Gaussians, $F_{sem}^q$ denotes the semantic feature map corresponding to the $q$-th object in the semantic cache $\Omega$. The semantic-awareness regularization term encourages Gaussians of the same category to share similar scales, thereby reducing computational overhead caused by redundant scales. After optimizing the

Table 1: Comparison results on ScanNet [6]. The best results are highlighted in **bold**, and the second-best results are underscored. "∗" indicates the use of the colmap-estimated poses following [52, 31, 32]. "−" indicates that the method does not support the specified task. "Rec., Seg., Bbox., Occ." denotes four multi-task evaluations: reconstruction quality, instance segmentation, 3D bounding box estimation, and semantic occupancy estimation.

| Tasks: | | | | Rec. | | Seg. | | Bbox. | | Occ. | |
|---|---|---|---|---|---|---|---|---|---|---|---|
| Method | Input | Online | Pose-free | PSNR | SSIM | mIoU | mAcc | AP | mAP | IoU | mIoU |
| LangSplat [31] | RGB | ✗ | ✗ | 18.4 | 0.69 | 27.5 | 51.3 | - | - | - | - |
| GaussianGrouping [52] | RGB | ✗ | ✗ | 19.6 | 0.74 | 32.6 | 56.9 | 43.6 | 24.5 | 47.4 | 22.1 |
| FeatureGS [32] | RGB | ✗ | ✗ | 23.9 | 0.84 | 41.1 | 66.0 | 51.4 | 32.7 | 50.9 | 31.2 |
| OpenGaussian [44] | RGB | ✗ | ✗ | 22.1 | 0.80 | 35.4 | 61.7 | 47.5 | 28.2 | 49.1 | 25.3 |
| InstanceGaussian [18] | Points | ✗ | ✗ | 24.5 | 0.83 | 40.5 | 65.7 | 52.3 | 33.4 | 53.5 | 32.8 |
| OpenScene [30] | Points | ✓ | ✗ | - | - | 42.8 | 68.6 | 55.7 | 34.8 | 51.8 | 30.5 |
| EmbodiedSAM [47] | RGB-D | ✓ | ✗ | - | - | 44.2 | 71.4 | 58.1 | 39.5 | 55.2 | 33.0 |
| SAM3D [49] | Points | ✓ | ✗ | - | - | 39.2 | 62.3 | 53.7 | 29.1 | 53.3 | 26.7 |
| Enhanced Baselines: | | | | | | | | | | | |
| HiCOM [10]+VFM [35] | RGB | ✓ | ✗ | 22.6 | 0.82 | 34.8 | 61.9 | 52.5 | 23.8 | 42.4 | 27.9 |
| MonoGS [26]+VFM [35] | RGB | ✓ | ✗ | 24.3 | 0.85 | 36.3 | 60.5 | 51.7 | 27.7 | 44.5 | 27.2 |
| EmbodiedOcc [45]+$\mathcal{L}_{RGB}$ | RGB | ✓ | ✗ | 17.6 | 0.65 | 29.2 | 54.8 | 56.2 | 35.6 | 54.6 | 33.1 |
| FeatureGS [32]+HiCOM [26] | RGB | ✓ | ✗ | 24.5 | 0.85 | 40.8 | 66.3 | 55.8 | 34.7 | 50.7 | 31.4 |
| **EA3D**∗ | RGB | ✓ | ✗ | 25.5 | 0.87 | 45.9 | 71.2 | **59.2** | 39.6 | 55.0 | **34.3** |
| **EA3D** | RGB | ✓ | ✓ | **25.8** | **0.89** | **46.3** | **71.8** | 57.9 | **39.9** | **55.4** | 33.9 |

integrated Gaussian features, we perform alpha-blending to accumulate the final splatted feature $\hat{F}$:

$$\hat{F} = \sum_{i \in N} F_i \cdot \alpha_i \prod_{j=1}^{i-1} (1 - \alpha_j), \tag{4}$$

where $\alpha_i$ denotes the opacity, $F_i$ is the integrated feature map of the $i$-th Gaussian.

**Joint Semantic-geometry Optimization.** During online Gaussian training, we jointly optimize Gaussian features and camera poses using a combination of photometric loss, geometric loss, knowledge-integrated loss, and regularization terms, formulated as:

$$\mathcal{L} = \sum_{t=0}^{t_{now}} \lambda_1 \mathcal{L}_1 + \lambda_2 \mathcal{L}_d + \lambda_3 \mathcal{L}_{kw} + \mathcal{L}_\delta, \tag{5}$$

where $\mathcal{L}_1$ is the $L_1$ photometric loss. $\mathcal{L}_d = \sum |\hat{D}_t - D_t|$, where $\hat{D}_t$ denotes the rendered depth from Gaussian splatting. $\mathcal{L}_{kw}$ denotes the $L_2$ distance between knowledge-integrated feature map and rendered feature map. $\lambda_1$, $\lambda_2$, and $\lambda_3$ are the weighting factors to balance the loss terms. $t_{now}$ denotes the current time step and $t_0$ is the initial frame. The loss is dynamically computed on the current frame to update existing Gaussian parameters and features, while future frames remain unseen.

## 4 Experiments

**Datasets.** We evaluate our method on two benchmarks: LERF [16] dataset comprises in-the-wild scenarios captured with the iPhone App Polycam. The objects in LERF include both common and long-tail categories with different sizes. Scannet [6] is an indoor dataset comprising each annotated with instance-level segmentation and labels across 200 categories. We use 10 RGB sequences selected by [30] without using the depth ground truth or any human annotations.

**Implementation Details.** We implement EA3D based on HiCoM with a fixed $\lambda_1 = 0.25$, $\lambda_2 = 0.1$, and $\lambda_3 = 0.15$. Each incoming frame is optimized with 100 motion steps, plus another 100 steps after adding new Gaussians. Every fifth frame is used as a test view. All training and testing data remain unseen to the off-the-shelf pretrained models to ensure a fair evaluation. All experiments are conducted on a single A100 80GB GPU. For more details, please refer to the Appendix.

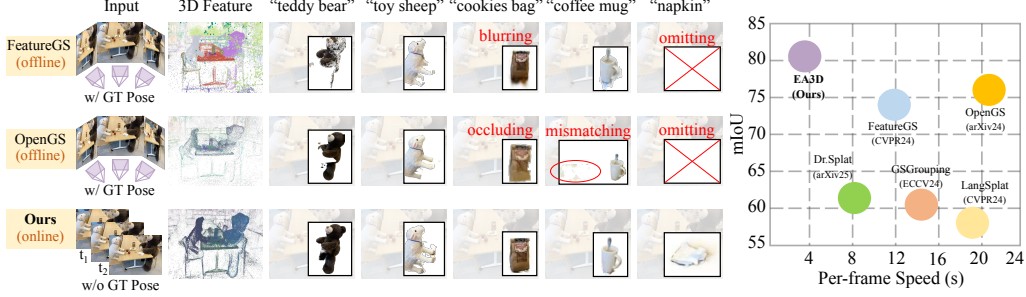

(a) Comparison of 3D objects extraction quality      (b) Comparison of quality and training time

Figure 4: **Visualization performance and model efficiency comparison with state-of-the-art methods.** Left (a): Under the more challenging streaming setting without pose input, EA3D delivers high-quality 3D object reconstruction and rendering. Notably, our method avoids redundant Gaussian features through efficient online updates, enabling more precise and lightweight optimization. Right (b): EA3D strikes a balance between speed and quality, significantly reducing training time while maintaining high-performance scene understanding.

Table 2: Comparisons under sparse views and online incremental settings on LeRF [16]. The best results are highlighted in **bold**, and the second-best results are underscored. "−" indicates methods do not support the specified task. "colmap" denotes offline pose estimation using COLMAP, "self." refers to online self-estimated poses. "Speed" denotes the average per-frame optimization speed.

| | Tasks: | | | Rec.(PSNR ↑) | | | Seg.(mIoU ↑) | | |
|---|---|---|---|---|---|---|---|---|---|
| Method | Online | Pose | Speed.(FPS) | 10 views | 30 views | 70 views | 10 views | 30 views | 70 views |
| LangSplat [31] | ✗ | colmap | 0.007 | 11.3 | 14.4 | 17.8 | 28.6 | 34.4 | 51.5 |
| FeatureGS [32] | ✗ | colmap | 0.018 | 15.2 | 18.9 | 22.4 | 29.4 | 41.2 | 53.6 |
| OpenGaussian [44] | ✗ | colmap | 0.005 | 14.9 | 19.5 | 22.7 | 30.1 | 40.5 | 55.8 |
| Enhanced Baselines: | | | | | | | | | |
| Cut3R [41]+VFM [35] | ✓ | self. | **0.648** | - | - | - | 33.7 | 26.5 | 21.9 |
| HiCOM [10]+VFM [35] | ✓ | colmap | 0.102 | 18.1 | 18.6 | 21.5 | 36.1 | 39.3 | 43.3 |
| **EA3D** | ✓ | self. | 0.235 | **21.9** | **21.8** | **23.2** | **53.8** | **55.0** | **57.4** |

## 4.1 Quantitative and Qualitative Comparisons

Our method enables holistic 3D object extraction across diverse tasks, including photo-realistic rendering, instance segmentation, and geometric reasoning (e.g., 3D bounding boxes, semantic occupancy, 3D mesh). We validate the effectiveness of our method through comparisons with state-of-the-art approaches and enhanced baselines in 3D reconstruction and online perception.

**Compared with reconstruction-based understanding methods.** We compare EA3D with NeRF-based [16] and Gaussian-based [31, 52, 44, 32, 18] approaches for 3D scene reconstruction with understanding. These methods rely on offline training with access to all scene views as input. Notably, the compared baselines also require camera poses from GT or Colmap estimated. For fair comparison, we incrementally replace our estimated poses with those from Colmap (denoted as EA3D*).

Results across multiple specific tasks are presented in Table 1. [52, 18, 31] utilize 2D semantic decoded by SAM as supervisions. While effective in 2D segmentation, this strategy fails to learn continuous 3D semantic-geometric representations. Our primary competitors [32, 44] incorporate semantic features but suffer from excessive redundant Gaussians and fail to achieve efficient joint convergence of geometry and semantics. Moreover, all the aforementioned methods rely on complete prior observations of the 3D space, which severely limits their applicability in real-world scenes. In contrast, EA3D adopts an online training strategy that delivers high-quality reconstruction and understanding, while offering better scalability.

**Compared with online 3D scene understanding methods.** Two common limitations can be observed across these approaches: 1) reliance on predefined geometry or 3D representations (e.g.,

point clouds, depth maps, meshes); 2) dependence on extensive training with large-scale annotated datasets. As shown in Table 1, our method achieves competitive performance even when compared to models trained specifically for the 3D understanding tasks. [49, 30, 47] utilize SAM to obtain 2D segmentations and project them into 3D space, but suffer from semantic ambiguities and multi-view inconsistency caused by mis-projections. In contrast, our approach jointly optimizes geometry and knowledge without relying on 3D priors, demonstrating the strengths of our unified online framework.

**Compared with enhanced baselines.** Since our work is the first to enable online joint geometry reconstruction and scene understanding, we enhance existing methods in two ways to serve as stronger baselines: 1) augmenting online reconstruction methods with scene understanding capabilities (e.g., HiCOM+VFM, MonoGS+VFM); 2) enabling online optimization of feature Gaussians (e.g., FeatureGS+HiCOM). Additionally, we incorporate an $L_1$ RGB loss into EmbodiedOcc [45], which was originally designed for online occupancy prediction. Table 1 demonstrates that EA3D consistently outperforms our baseline HiCOM by integrating VFM-driven scene understanding. It also surpasses FeatureGS+HiCOM, which similarly employs semantic features and online updates, highlighting the effectiveness of our unified framework. Furthermore, compared to online SLAM-based methods [45, 26], EA3D achieves better results in both geometric reconstruction and scene interpretation.

**Qualitative Comparisons.** We further compare the visual quality of 3D object extraction with the baseline methods in Fig. 4(a). Given a streaming video without pose information, EA3D allows high-quality reconstruction and rendering of arbitrary 3D objects. Visualizations of the 3D features show that our online feature Gaussians efficiently and accurately capture both geometry and semantics. In contrast, leading baselines introduce redundant noise, produce inferior renderings, and fail to extract challenging objects (e.g., a small piece of napkin). EA3D also enables a variety of downstream applications, such as manipulation simulation, motion emulation, controllable 3D editing, and object insertion or removal. Additional results and applications are presented in the Appendix.

Our experimental results and theoretical analyses reveal that naïve integrations of existing models tend to perform poorly and may even degrade overall performance due to inherent conflicts among components. In contrast, our method fully harnesses the open-vocabulary features extracted by VFMs and effectively tackles the key challenges of 3D semantic consistency and online geometric reconstruction. Moreover, it achieves higher efficiency and lower computational overhead through a unified and elegantly designed framework.

## 4.2 Sparse Views and Online Stability

Table 2 reports the performance and robustness of EA3D under sparse-view and online incremental settings. We evaluate it by sequentially inputting sparse-view images (e.g., 10 views) and progressively extending the sequence length. In contrast, offline baselines [31, 32, 44] receive all training views at once. Results show that our method exhibits strong robustness to sparse-view inputs, achieving promising results even with a few initial frames in the early stage. As the sequence length increases ($10 \rightarrow 30 \rightarrow 70$ views), EA3D maintains stable quality, while baseline methods struggle with instability and slow convergence under sparse inputs. Fig. 3 further illustrates the online updating process of rendering and segmentation, occupancy estimation, and 3D mesh generation with EA3D.

Table 3: Ablation on key components. "Train" and "Render" represent the per-frame training and rendering time, measured in FPS. "regular.term" denotes the semantic-awareness regularization. "online.opt", "online.odo", and "joint.opt" denote the online updating strategy, online visual odometry, and joint optimization, respectively.

| Strategy | PSNR | mIoU | mAcc | Train | Render |
|---|---|---|---|---|---|
| Baseline: HiCoM [10] | 22.6 | 34.8 | 61.9 | 0.29 | 230 |
| W/o CLIP Encoder | 25.3 | 41.6 | 66.4 | 0.28 | 220 |
| W/o SAM Encoder | 25.4 | 42.8 | 67.1 | 0.27 | 215 |
| W/o regular.term | 25.1 | 44.3 | 70.5 | 0.21 | 208 |
| W/o online.opt | 24.6 | 44.5 | 69.7 | 0.07 | 110 |
| W/o online.odo | 25.0 | 45.4 | 70.8 | 0.26 | 205 |
| W/o joint.opt | 24.8 | 45.7 | 71.4 | 0.25 | 210 |
| **Ours-full** | 25.8 | 46.3 | 71.8 | 0.23 | 210 |

### 4.3 Model Efficiency Analysis

Our method enables online incremental reconstruction and understanding of scenes for 3D object extraction. Here, we quantitatively evaluate the speed and memory usage of each key component. As shown in Fig. 4(b), our method achieves faster optimization while maintaining top performance. EA3D strikes a balance between speed and accuracy, delivering higher rendering efficiency with reduced storage overhead. Detailed quantitative experimental results are provided in the Appendix.

### 4.4 Ablation Studies

As shown in Table 3, we conduct ablation studies and analyze the key components of our designs for online open-world 3D object extraction. Embedded visual features from VFMs (e.g., CLIP [9] and SAM [33]) imbue Gaussians with semantic awareness, enhancing both fine-grained geometry modeling and scene understanding. Our online optimization strategy accelerates feature Gaussian refinement via an efficient feedforward mechanism, ensuring accuracy while minimizing redundancy. The online visual odometry provides dynamic pose updates and dense geometric cues, speeding up convergence. Semantic-aware regularization links Gaussian geometry with semantic features, ensuring object-level 3D consistency and smoothness. By jointly optimizing geometry, semantics, and pose, our method enables recurrent feature updates that seamlessly integrate appearance and structure for robust 3D reconstruction and understanding. For more ablation studies on key modules and hyperparameters, please refer to the Appendix.

## 5  Conclusion

We have presented EA3D, a unified online framework for open-world 3D object extraction. EA3D enables simultaneous online reconstruction and understanding without geometric or pose priors. It consistently achieves good performance across a broad set of tasks, including photo-realistic reconstruction and rendering, semantic and instance segmentation, 3D bounding box construction, semantic occupancy estimation, and 3D mesh generation. EA3D introduces a novel perspective for aligning and aggregating 3D semantic and geometric features through online reconstruction and dynamic update strategies. It establishes a unified online 3D feature aggregation framework grounded in reconstruction constraints, enabling more accurate and efficient 3D scene understanding and reconstruction.

## Acknowledgment

This work was supported by National Key R&D Program of China (Grant No. 2022ZD0160305). This work was also a research achievement of Key Laboratory of Science, Technology, and Standard in Press Industry (Key Laboratory of Intelligent Press Media Technology). Ming-Hsuan Yang was supported in part by the Institute of Information & Communications Technology Planning & Evaluation (IITP) grant funded by the Korean Government (MSIT) (No. RS-2024-00457882, National AI Research Lab Project).

## Broader Impacts

This paper presents research aimed at advancing the fields of 3D vision, which hold significant promise for enhancing the 3D object extraction. While AI-driven scene reconstruction and perception bring benefits, they could also raise concerns regarding their social and economic impacts. Automating 3D labeling and perception tasks can potentially disrupt the labor market, posing risks to certain job sectors, particularly in sectors that rely on manual data annotation. It is crucial to exercise caution and ensure that the societal implications are thoroughly addressed.

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
