# OpenReview forum: "EA3D: Online Open-World 3D Object Extraction from Streaming Videos"
_NeurIPS.cc/2025/Conference — NeurIPS 2025 poster_

### Official Review · Reviewer_Fu59 · 2025-07-01

**Clarity:** 3
**Significance:** 2
**Originality:** 2
**Rating:** 4
**Confidence:** 4

**Summary:**

The paper presents ExtractAnything3D (EA3D), an online framework for open-world 3D object extraction from streaming video. It integrates semantic understanding and geometric reconstruction in real time without requiring pre-built 3D models or global poses. By combining vision-language models and visual foundation models, EA3D incrementally builds a knowledge-guided 3D Gaussian representation. A joint optimization module updates geometry and semantics simultaneously. The method supports various tasks like rendering, segmentation, and 3D object detection, and achieves strong results across several benchmarks.

**Questions:**

1)How is the semantic accuracy of VLMs evaluated across complex or occluded scenes? “We use the prompt ‘Find and list all the possible objects in the given image’...” (Line 122). While this prompt is general, have the authors measured how robust the VLM is under occlusion, motion blur, or low-resolution input?

2)How does the online semantic cache handle ambiguous or overlapping object categories? Line 124, Is there a mechanism to merge semantically similar objects (e.g., “mug” vs. “cup”)? How is category drift avoided over time?

3)Given the variability in language models’ responses, is there a confidence threshold or filtering strategy to ensure physical attributes are meaningful and consistent?

4)On Feature Embedding & Gaussian Integration, What are the parameters (e.g., number of neighbors) used for aggregation? Is the method robust across varying object scales and densities? like Line 134

**Ethical Concerns:**

["NO or VERY MINOR ethics concerns only"]

**Final Justification:**

This is an excellent paper that makes good contribution to the field. I enjoyed reading it and congratulate the authors on their fine work. I recommend its acceptance.

**Limitations:**

1）While the method claims to handle occluded regions through "novel view synthesis," there is little quantitative or qualitative analysis of how well semantic coherence is preserved under severe occlusions

2）The paper proposes online visual odometry for camera pose estimation, but does not provide sufficient evaluation or discussion of its accuracy or robustness under challenging conditions (e.g., low texture, fast motion)

**Paper Formatting Concerns:**

Some typos can be corrected later

**Quality:**

3

**Strengths And Weaknesses:**

Strengths：

1）The method enables real-time 3D scene understanding and reconstruction without relying on offline-collected multi-view data or precomputed geometry. “...without geometric or pose priors” (line 52)

2）EA3D dynamically interprets each frame using vision-language models (e.g., CLIP) and visual foundation models, enabling generalizable object-level understanding in diverse environments.

Weaknesses：

1）Although the method runs online, using multiple large-scale foundation models (VLMs, VFMs) per frame could lead to latency or hardware bottlenecks in real-world deployments

2）As the system maintains a growing knowledge-integrated Gaussian map and historical feature cache, it’s unclear how memory usage and performance scale over time or in large-scale environments

---

> ### Author Rebuttal · Authors · 2025-07-31
>
> We sincerely appreciate your thoughtful comments. We have carefully considered each of your questions and provided detailed responses below. We will include these discussions in the revised paper.
> ***
> ### [W1. Latency and hardware bottlenecks]
> Unlike existing methods, our approach does not rely on explicit decoding from vision foundation models. Instead, we directly encode features extracted from the backbone, which significantly reduces the latency and hardware demands associated with large-scale foundation models. Meanwhile, our online synchronization process ensures high parallel computation efficiency throughout the pipeline. As shown in Tab II (Appendix) and Tab 2 (main text), EA3D strikes a balance between speed and accuracy compared with baseline methods using VLMs and VFMs. As shown below, EA3D also delivers higher quality, fewer parameters, and reduced storage overhead.
>
> | Metrics |Definitions| Baseline (GSGrouping) | **Ours** |
> | :-|:-|:-:|:-:|
> | Latency (FPS) $\uparrow$ | Foundation Models (VFMs) Embedding | 0.19 | **0.43** |
> | Latency (FPS) $\uparrow$ | Online Visual Odometry | 0.49 | **1.67** |
> | Latency (FPS) $\uparrow$ | Gaussian Training | 0.13 | **0.84** |
> | Latency (FPS) $\uparrow$ | Total Speed | 0.06 | **0.23** |
> | PSNR $\uparrow$ | Reconstruction Quality | 19.6 | **25.8** |
> | Parameters (M) $\downarrow$|  Model Size |460 | **364** |
> | Hardware| Experimental Setup | single A100 | single A100 |
>
> For fair comparisons, all experiments were conducted using the same device. We acknowledge that using large-scale foundation models may introduce latency and hardware bottlenecks in real-world deployments. In future work, we will explore further lightweight computation to make our method more suitable for deployment. We will provide a clearer discussion of the potential latency and hardware limitations in the real world in the revised version.
> ***
> ### [W2. Memory usage and performance]
> Our semantic-aware adaptive Gaussian, together with its online updating strategy, ensures robust performance and efficiency as time and scene scale increase. Table 2 illustrates the robustness of our method as temporal and spatial scales increase. The table below provides more detailed results, quantitatively validating model performance and memory usage under these expanding conditions.
>
> |Timestep(/frame) | Reconstruction (PSNR) $\uparrow$  | Segmentation (mIoU) $\uparrow$ |Memory Usage (G)|
> |:-:|:-:|:-:|:-:|
> | 10 | 23.5 | 43.2 | 30 |
> | 100 | 24.4 | 44.3 | 36 |
> | 500 | 25.0 | 45.1 | 39 |
> | 2000 | 25.3 | 45.5 | 45 |
>
> Although large-scale scenes are not the primary focus of our target applications, we still evaluate EA3D's online geometric reconstruction and scene understanding capabilities in different scene scales, as shown below:
>
> |3D Scale ($m^3$)|Reconstruction (PSNR) $\uparrow$ |Segmentation (mIoU) $\uparrow$|Memory Usage (G)|
> |:-:| :-: |:-:|:-:|
> |40|28.7|49.4|43|
> |90| 28.3 | 47.6 | 52 |
> |260| 26.5 | 48.1 | 67 |
>
> The results demonstrate that even under significant temporal and spatial growth, our method maintains good performance without incurring notable increases in computational cost or memory usage.
> ***
> ### [Q1. Robustness of the foundation models]
> The robustness of the VLM and VFM under challenging conditions—such as occlusion, motion blur, or low texture—is not our primary focus. Instead, our method emphasizes fundamental feature alignment based on multi-view reconstruction, with a core objective of ensuring consistency in 3D geometry and features. Moreover, there exists no reliable benchmark for evaluating the severity of occlusion, rapid motion, or low texture in 3D scenes, making it difficult to provide quantitative results. We manually selected several challenging scene segments, each featuring significant occlusion, fast camera motion, or low-resolution images obtained via downsampling. As shown in the table below, although VLM-VFM struggles under such challenging conditions, our method achieves better 3D consistency across geometry and semantics, benefiting from multi-view geometry and 3D feature alignment.
>
> |Method|Occlusion (mIoU)|Fast Motion (mIoU)|Low-resolution (mIoU)|
> |:-|:-:|:-:|:-:|
> |VLM-VFMs|27.8|29.7|32.3|
> |**Ours**|**43.2**|**40.8**|**41.5**|
>
> EA3D helps mitigate the semantic ambiguity and missing information of VLMs and VFMs in these cases, providing stronger feature alignment for the foundational model features through 3D geometric constraints and a dynamic online updating strategy.
>
> Nevertheless, as VLM-VFM models continue to advance, these limitations are expected to be gradually addressed, providing stronger priors for our method and enabling further performance gains. More qualitative results will be added in our revised paper.
> ***
> ### [Q2. Handling semantic ambiguity and overlapping]
> We address the challenges of semantic ambiguity and overlap through instance-level feature selection and aggregation, combined with a semantic updating threshold. We first mitigate semantic feature conflicts of the same object across different views through instance-level semantic feature map extraction (Lines 129–138). Each object instance is assigned a unique ID along with its aggregated semantic feature during the online reconstruction, while the highest-confidence semantic is associated, effectively avoiding conflicts among multiple semantic predictions. We employ pretrained visual foundation models to obtain instance-level features, where category similarity is computed between semantic features and embedded continuous vectors. We then aggregate instance-level features online to assign a unique ID to each object, selecting and fusing the most confident semantic feature in the current view. This explicitly aligns instance-level semantics and effectively resolves ambiguity.
>
> We also introduce an updating threshold (Appendix Lines 175–182) to dynamically update the semantic cache, allowing only high-confidence semantics to replace existing entries. As shown in Table IV, our method demonstrates strong robustness across varying thresholds, with ambiguous semantics implicitly corrected through multi-view reconstruction.
> ***
> ### [Q3. Physical attributes filtering]
> Yes, we apply an aggregation strategy together with an updating threshold for maintaining the consistency of each instance's physical attributes, just as we do for semantic categories. Similar to the semantic, each object's physical attributes are also associated with a unique instance ID and its aggregated feature (Lines 142-143). During online scene reconstruction, physical attributes with higher confidence and frequency are given greater weight and assigned to the object as its definitive properties. A similar updating threshold (Appendix Lines 175–182) for physical attributes is assigned to dynamically update the semantic cache and the corresponding physical attributes in it, while inconsistent attributes are filtered out during the online updating process.
> ***
> ### [Q4. Parameters for aggregation and robustness]
> We ensure the robustness of our method to variations in object scales and densities from two aspects: 1) similarity-based K representative features; and 2) 3D geometric constraints guided by semantic-aware adaptive Gaussians.
>
> We adopt the K-Means algorithm on associated features (following Ovrir-3D$^{[2]}$) to represent each instance. Each 3D instance is ranked based on the maximum cosine similarity between the semantic cache query and its K representative features. We further provide an ablation study on the hyperparameter $K$, demonstrating that this strategy more effectively aggregates representative features across multiple viewpoints and offers greater robustness to variations in object density and scale.
>
> |$K$|PSNR $\uparrow$|mIoU $\uparrow$|
> |:-:|:-:|:-:|
> |16|24.7|45.4|
> |32|25.3|45.7|
> |64|25.8|46.3|
>
> Leveraging semantic-aware adaptive Gaussians, we also enable the individual reconstruction of instances based on their clustering center features, ensuring accurate geometric alignment across objects of varying scales. This strategy enhances robustness to object scale and density in 3D space through multi-view geometric constraints at consistent spatial viewpoints.
> > [2] OVIR-3D: Open-Vocabulary 3D Instance Retrieval Without Training on 3D Data.
> ***
> ### [L1&L2. Performance under challenging conditions]
> Measuring the precise level of occlusion, low texture, and motion speed in existing datasets is a non-trivial challenge, and we recognize this as a limitation of the evaluation methodology. To further quantify the robustness and accuracy of our model under various challenging conditions, we manually selected scenes with severe occlusion, fast camera motion, and simulated low-texture environments using a downsampling algorithm. We conducted targeted validation experiments on these scenes, as shown in the table below. ''Reconstruction'' reflects the accuracy of online geometry and visual odometry, while ''Semantics'' represents multi-view 3D understanding, which can be used to assess semantic coherence.
>
> |Scene|Scene001 (occlusion+fast motion) |Scene002 (low texture+fast motion) | Scene003 (occlusion+low texture) |
> |:-|:-|:-|:-|
> |Baseline (Reconstruction)|17.5|20.3|21.1|
> |**Ours (Reconstruction)**|**22.8**|**23.9**|**24.6**|
> |Baseline (Semantic)|31.8|35.6|33.0|
> |**Ours (Semantic)**|**39.2**|**41.4**|**42.7**|
>
> Moreover, we kindly suggest that the qualitative results in our supplementary material, particularly in the video demonstrations, provide compelling visual evidence that our "feature 3D alignment" mechanism helps maintain semantic consistency even in challenging, partially occluded, and fast motion scenarios. We agree that a dedicated quantitative metric for this phenomenon would be a valuable addition and an interesting direction for future work. More discussion will be added in our revised paper.
> ***
> If you have any further questions/concerns, please do not hesitate to let us know.
>
> Thank you very much,
>
> Authors

---

> > ### Author Response · Authors · 2025-08-02
> >
> > Dear Reviewer,
> >
> > We appreciate the time and effort you've dedicated to reviewing our submission. We have carefully addressed your comments in our response and would be glad to provide any further clarification before the discussion period ends (August 6, 11:59 pm AoE). We hope that you can consider raising the score after we address all the concerns.
> >
> > Thank you again for your time, and look forward to your replies.
> >
> > Sincerely, authors of submission 5333

---

> > ### Comment · Reviewer_Fu59 · 2025-08-04
> > **Response to 5333 Authors' Rebuttal**
> >
> > Thank you for your detailed rebuttal. I appreciate the additional experiments and clarifications provided. Below are my responses to your answers:
> >
> > **W1**: Your performance comparison table is convincing and demonstrates clear improvements over baseline methods. The acknowledgment of real-world deployment limitations and commitment to explore lightweight computation in future work is reasonable.
> >
> > **W2**: The quantitative results showing stable performance across different temporal and spatial scales effectively address my concerns. The memory usage growth appears manageable, and the performance metrics remain consistent.
> >
> > **Q2** : Your explanation of instance-level feature selection and the updating threshold mechanism adequately addresses how overlapping categories are handled. The unique ID assignment strategy is a reasonable solution.
> >
> > **Q4** : The ablation study on K values (16, 32, 64) provides useful insights into parameter sensitivity and demonstrates robustness across different settings.
> >
> > I think these issues have been addressed well.
> >
> > **Q1** : While I appreciate the additional experiments, your response somewhat sidesteps the core question. Stating that VLM robustness is "not our primary focus" is concerning since your method heavily relies on VLMs for semantic understanding. The manually selected challenging scenes may introduce selection bias. More importantly, you haven't directly addressed how the generic prompt "Find and list all the possible objects" performs under the specific challenging conditions I mentioned. Consider providing a more systematic evaluation methodology or at least acknowledge this as a significant limitation that could affect real-world deployment.
> >
> > **L1&L2**: Your acknowledgment that measuring occlusion levels and motion speed is "non-trivial" is honest, but this limitation significantly impacts the evaluation credibility. The manually selected scenes approach is problematic due to potential selection bias.
> >
> > **Q3**: Your response lacks specificity regarding confidence thresholds and filtering strategies. While you mention an "updating threshold," you don't provide quantitative results or specific parameter values that would help readers understand the method's reliability.
> >
> > These questions didn't solve my doubts very well.
> >
> > I still think it's a great paper, the rebuttal has addressed most major concerns adequately. I maintain my rating but encourage the authors to incorporate the discussed limitations and additional results in the final version.

---

> ### Author Response · Authors · 2025-08-05
> **Response to Reviewer Fu59**
>
> Dear Reviewer Fu59,
>
> We sincerely thank you for the constructive feedback and recognition of our work. We further address your concerns as follows.
>
> **Q1**: We acknowledge the importance of challenging scenes in real-world deployments and agree that current 3D scene benchmarks often overlook the need for specifically designed validation sets for these challenging scenes. For the generic prompt "Find and list all the possible objects", we have conducted extensive experiments across over 1,500 scenes and more than 2.5 million views, which include various challenging conditions as mentioned. While our method demonstrates strong performance across diverse scenes and settings, we recognize that the lack of targeted quantitative evaluations and diverse challenging scenarios hinders deeper analysis of VLM robustness and prompt effectiveness. The experiments provided in our rebuttal include six selected challenging scenes, covering 24 clips with hard cases such as severe occlusion, fast motion, and low resolution. This data was carefully chosen to be difficult, not to be biased toward a certain outcome. Due to the time constraints of the rebuttal phase, we were unable to exhaustively identify all extreme scenarios within the benchmark. However, we are committed to completing the full statistics and providing comprehensive experiments and analysis in the camera-ready version. We greatly appreciate your insightful suggestion and believe it will help advance the field.
>
> **L1&L2**: We acknowledge this as a valid concern and will aim to mitigate the bias by developing more effective evaluation techniques and benchmarks tailored to challenging scenarios. We will also further discuss and highlight the corresponding limitations and solutions in the revised paper. Inspired by your insightful suggestion, we believe extending 3D objects extraction to more challenging scenes is an underexplored yet crucial direction in 3D scene understanding, and we will pursue it as part of our future work.
>
> **Q3**: We would like to clarify what we meant in the response. We first apply an aggregation strategy for physical attributes through instance-level feature map fusion, which is performed during the online cache update. In this process, the physical attribute features with the highest occurrence frequency and confidence are dynamically fused as a variable-length vector $Y$ into the online semantic cache (Lines 142–143), constrained by multi-view 3D consistency. The physical attribute feature $Y$ is then integrated into the feature map $F$ for semantic cache. The semantic cache updating threshold (Lines 218-221 in the appendix) is thus applied on the feature map $F$ and is shared for both embedded semantic features and physical attribute features (Lines 152–157). Given the inherent correlation between semantic and physical attribute features, the use of a unified semantic cache updating threshold for both ensures more coherent knowledge feature fusion and consistent robustness. Thus, the ablation on the threshold hyperparameters in Table IV also demonstrates the robustness of physical attributes under different settings (the same as Q2). We will clarify this further in the camera-ready version.
>
> We hope these clarifications have addressed your concerns. We are happy to discuss any aspects of the paper that may require further explanation. We also hope that this will resolve any remaining issues and encourage you to raise the rating. Thank you again for your valuable and insightful comments.
>
> Best regards.

---

### Official Review · Reviewer_NWKd · 2025-07-02

**Clarity:** 2
**Significance:** 2
**Originality:** 2
**Rating:** 4
**Confidence:** 3

**Summary:**

This paper presents ExtractAnything3D (EA3D), a unified framework for online 3D scene understanding in open-world settings. The idea is to let a system explore and understand a new environment on the fly—just like a robot walking into an unfamiliar room and making sense of everything it sees, without relying on pre-built 3D maps or multi-view data.

EA3D takes streaming video input and uses vision-language models and visual foundation models to extract object-level knowledge in real time. This information is integrated into a feature map built with Gaussian representations, which gets updated as more frames come in. A recurrent optimization module helps focus on important areas, improving both geometry and semantics. The system can handle tasks like 3D reconstruction, semantic segmentation, and even physical reasoning, all while running online.

**Questions:**

The paper lacks comparisons or discussions with several relevant baselines. Please consider adding these to better situate the contribution.

The framework is quite complex and integrates many existing components. I look forward to hearing other reviewers' perspectives and the authors' response regarding the core novelty.

**Ethical Concerns:**

["NO or VERY MINOR ethics concerns only"]

**Final Justification:**

The rebuttal has resolved my concerns. I have also read the reviews and rebuttals from the other reviewers. Others also acknowledge the paper's contribution as a system-level solution, which I agree with. Therefore, I will maintain my current rating. However, I encourage the authors to include additional empirical discussions in the camera-ready version, particularly regarding methods that were not compared in the current submission, such as VLM-3R, UniForward, and PanSt3R.

**Limitations:**

yes

**Quality:**

3

**Strengths And Weaknesses:**

## Strengths

### Practical relevance:
EA3D tackles an important and challenging problem—online open-world 3D scene understanding. The system can dynamically extract, reconstruct, and interpret arbitrary 3D objects from streaming video, without requiring known geometry, camera poses, or predefined object categories. This makes it well-suited for real-world robotic and embodied AI applications.

### Comprehensive capability:
The proposed framework unifies multiple tasks—including geometry reconstruction, semantic/instance segmentation, and physical reasoning.

### Strong empirical results:
The method is evaluated on a wide range of benchmarks and tasks (e.g., photorealistic rendering, 3D mesh generation, semantic occupancy estimation), showing strong performance across the board.The authors commit to releasing the code, which is helpful for the community.

## Weaknesses

### Lack of comparison with key baselines:
The paper misses comparisons and/or empirical discussions with several relevant and recent works in both (i) reconstruction + segmentation (e.g., Feature4X, UniForward, Large Spatial Model, PanSt3R, SpatialSplat, SAB3R) and (ii) reconstruction + VLM (e.g., VLM-3R, Spatial-MLLM, Learning from Videos for 3D World). These comparisons are crucial to situate the contribution.

### Limited clarity on novelty:
While the overall system is well-integrated and comprehensive, it's unclear what the core novelty is beyond combining existing components. Many of the ideas are present in prior works. A clearer articulation of what is truly new in EA3D would strengthen the contribution.

---

> ### Author Rebuttal · Authors · 2025-07-31
>
> We sincerely appreciate your thoughtful comments. We have carefully considered each of your questions and provided detailed responses below. We will include these discussions in the revised paper.
> ***
> ### [W1&Q1. Comparison with more baselines]
> Thank you for the constructive feedback. As shown in the table below, most of the papers mentioned above are arXiv submissions that remain unpublished as of the submission deadline of this conference. These works have also not released their code or testing details, and some tackle different tasks, making it difficult to evaluate them using the same metrics in 3D. We have reproduced the publicly available methods and made our best effort to replicate the most recent unpublished works (despite their release dates postdating our submission deadline) to further evaluate the effectiveness of our proposed approach. As presented in the table (* denotes our own reimplementation due to unavailable code), our method consistently demonstrates superior performance in both 3D geometric reconstruction and 3D scene understanding:
>
> | Method        |   Details   |          3D Reconstruction (PSNR) $\uparrow$ |    3D Segmentation (mIoU) $\uparrow$     |
> | :--------  |    :--------|  :--------:|      :--------:|
> | UniForward        |    Not open-sourced (arXiv 2025/06/11)   |          N/A |          N/A |
> | PanSt3R        |    Not open-sourced (arXiv 2025/06/26)   |          N/A |          N/A |
> | SAB3R        |    Not open-sourced (arXiv 2025/06/02)   |          N/A |          N/A |
> | VLM-3R        |    VQA model (arXiv 2025/06/01)   |          N/A |          N/A |
> | Spatial-MLLM        |    VQA model (arXiv 2025/05/29)   |          N/A |          N/A |
> | Large Spatial Model        |    NeurIPS 2024   |          24.33 |          40.21 |
> | Learning from Videos for 3D World|    (arXiv 2025/05/30)   |          N/A |          42.10 |
> | Feature4X*        |    Not open-sourced   |          21.75 |          38.26 |
> | SpatialSplat*        |    Not open-sourced (arXiv 2025/05/29)   |          23.86 |          41.53 |
> | **Ours**        |    Submission on 2025/05/15   |          **25.85** |          **46.34** |
>
> In the revised paper, we will include more comparisons with these methods and supplement the experimental results once their corresponding code is publicly released.
> ***
> ### [W2&Q2. Clarity on novelty]
> While EA3D does utilize established components, our core novelty is not in their combination, but in a new unified framework for online feature alignment and 3D aggregation within a dynamic, multi-view reconstruction process. The key innovation of EA3D lies in its ability to implicitly align imperfect semantic and geometric features from foundation models, continuously aggregating and refining them within 3D space as an integral part of the reconstruction process. Prior methods either naively concatenate features from separate 2D and 3D modules—which can lead to inconsistencies, ambiguity, and performance degradation—or perform feature aggregation in a disjointed, offline manner. In contrast, EA3D unifies these processes by leveraging multi-view geometric constraints to guide feature alignment in an end-to-end manner, seamlessly supporting both reconstruction and 3D scene understanding. This novel approach enables a more accurate and efficient online 3D scene interpretation than previously possible.
>
> Instead of treating reconstruction and understanding as separate tasks, our method integrates them into a single, cohesive framework, leading to a more robust and holistic understanding of the 3D scene. This strategy enables our method to achieve superior performance beyond existing modular combinations, reduces error accumulation, and ensures high efficiency and rapid updates in online settings. The table below highlights this contribution, demonstrating the crucial role of our novel unified design.
>
> |Components|3D Reconstruction (PSNR) $\uparrow$ | 3D Segmentation (mIoU) $\uparrow$ |Speed (fps) $\uparrow$     |
> | :-------- | :--------:|:--------:|:--------:|
> | 3DGS+VFM | 18.2 | 28.1 | 0.01 |
> | HiCOM+VFM | 22.6 | 34.8 | 0.10 |
> | Cut3R+VFM | N/A | 32.7 | 0.19 |
> | FeatureGS+Online-Pose+VFM | 24.5 | 40.8 | 0.03 |
> | MonoGS+VFM | 24.3 | 36.3 | 0.07 |
> | Baseline w/o features|19.8|33.5|0.04 |
> | Baseline w/o feature online 3D alignment|23.9|41.9|0.09 |
> | **Ours-unified**|**25.8**|**46.3** |**0.23** |
>
> EA3D presents a novel, effective solution that unifies 3D geometric reconstruction and scene understanding within a single online pipeline. "w/o features" represents a direct composition of modules via input–output connections. "w/o feature online 3D alignment" lacks our implicit 3D alignment. EA3D significantly outperforms these baselines in both performance and efficiency, yielding more consistent and high-quality 3D reconstructions. Please note that  "w/o features" is time-consuming because it needs to decode all features into 2D semantic maps, and "w/o feature online 3D alignment" relies on the offline feature-GS reconstruction. By introducing the online feature Gaussians representation coupled with joint visual odometry, EA3D efficiently utilizes past observations to reconstruct and interpret 3D scenes, achieving a strong balance between accuracy and speed.
>
> In Appendix Sections C & D (Lines 82–163), we provide a more detailed analysis of our method's advantages and comprehensive comparisons with existing baselines and their combinations. We also construct stronger baselines by combining current VLM, VFM, 3D reconstruction, and pose estimation methods. Results in Table 1 (main text) and Table I (Appendix) show that naive integrations of existing models perform poorly and can even degrade performance due to conflicts between components. In contrast, our method not only fully leverages the open-vocabulary features extracted by VFMs but also effectively addresses key challenges in 3D semantic consistency and online geometry reconstruction while improving efficiency and reducing computational overhead through a unified framework.
> ***
> If you have any further questions/concerns, please do not hesitate to let us know.
>
> Thank you very much,
>
> Authors

---

> > ### Author Response · Authors · 2025-08-02
> >
> > Dear Reviewer,
> >
> > We appreciate the time and effort you've dedicated to reviewing our submission. We have carefully addressed your comments in our response and would be glad to provide any further clarification before the discussion period ends (August 6, 11:59 pm AoE). We hope that you can consider raising the score after we address all the concerns.
> >
> > Thank you again for your time, and look forward to your replies.
> >
> > Sincerely, authors of submission 5333

---

> > > ### Comment · Reviewer_NWKd · 2025-08-06
> > >
> > > Thank you for the rebuttal — it has resolved my concerns. I have also read the reviews and rebuttals from the other reviewers. Others also acknowledge the paper's contribution as a system-level solution, which I agree with. Therefore, I will maintain my current rating.
> > >
> > > However, I encourage the authors to include additional empirical discussions in the camera-ready version, particularly regarding methods that were not compared in the current submission, such as VLM-3R, UniForward, and PanSt3R.

---

> ### Author Response · Authors · 2025-08-09
> **Response to Reviewer NWKd**
>
> Dear Reviewer NWKd,
>
> We sincerely thank you for acknowledging that our rebuttal has resolved your concerns and for recognizing the contribution of our work. We also appreciate your suggestion to include additional empirical discussions, particularly regarding methods such as VLM-3R, UniForward, and PanSt3R. We will incorporate these comparisons and analyses in the camera-ready version to further strengthen the paper.
>
> In addition to the system-level perspective, we would like to highlight that EA3D is designed as a unified and efficient framework for joint online 3D reconstruction and holistic scene understanding, aiming to overcome the inherent limitations of simply building a system by combining existing components. Our proposed techniques include knowledge feature map aggregation, online feature Gaussians, and a dynamic updating strategy, which are, to the best of our knowledge, novel and unexplored in prior works. By effectively filling a gap in the literature on online joint geometric reconstruction and holistic scene understanding, our method delivers a state-of-the-art solution to this challenging task.
>
> We hope that these contributions, alongside the planned additions in the camera-ready version, will fully meet your expectations and encourage you to raise the rating. Once again, thank you for your valuable guidance and for helping us improve this paper.
>
> Best regards.

---

### Official Review · Reviewer_ouYe · 2025-07-03

**Clarity:** 3
**Significance:** 3
**Originality:** 3
**Rating:** 4
**Confidence:** 4

**Summary:**

This paper proposes EA3D, a unified online open-world 3D object extraction framework that can simultaneously perform geometric reconstruction and overall scene understanding from streaming videos. The framework demonstrates effectiveness in various tasks, including photorealistic rendering, semantic and instance segmentation, and its advantages have been verified through experiments.

**Questions:**

1.If EA3D* uses an offline camera pose estimation model that is better than COLMAP, will it produce better results?

2.EA3D mentions matching distributions between the current frame and the previous frame, and using the matching distribution to propagate gs features. However, how is the new current frame feature calculated, and how is the gs feature updated?

3.Why does EA3D achieve better reconstruction performance, and which contribution of EA3D improves the reconstruction quality?

4.Do the reconstruction metrics in Tab1 refer to the performance of novel view synthesis?

5.Can EA3D be understood as a model that reconstructs 3DGS online while progressively updating online gs features?

6.Is the excellent performance of EA3D in downstream tasks due to more efficient feature fusion or more powerful VFM features?

**Ethical Concerns:**

["NO or VERY MINOR ethics concerns only"]

**Final Justification:**

Rebuttal has addressed some of my concerns. I agree that this is a contribution in terms of a strategy or a system. I have decided to maintain my score.

**Limitations:**

Yes

**Quality:**

3

**Strengths And Weaknesses:**

Strengths:

It does not require pre-constructed 3D geometry or camera pose information, and can directly process streaming videos, perform real-time scene reconstruction and understanding, and support incremental updates in dynamic environments.

The unified framework supports multiple 3D tasks, including photorealistic rendering, semantic/instance segmentation, 3D bounding box estimation, semantic occupancy prediction, and 3D mesh generation. Moreover, it outperforms existing offline and online methods in multiple benchmark tests.

Weakness:

The paper relies on existing modules to reconstruct 3D scenes.

---

> ### Author Rebuttal · Authors · 2025-07-31
>
> We sincerely appreciate your thoughtful comments. We have carefully considered each of your questions and provided detailed responses below. We will include these discussions in the revised paper.
> ***
> ### [W1. Differences from existing modules]
> Thank you for the constructive feedback. While EA3D utilizes existing modules as building blocks, it requires the proposed algorithmic design and novel components to achieve state-of-the-art results. The core contribution lies in establishing a unified, efficient framework for online 3D reconstruction and holistic scene understanding, which addresses the critical limitations of simply combining existing models. Tables 1 and 2 show that our method significantly outperforms the baselines built on combinations of existing modules across all evaluation metrics. We clarify and emphasize the contributions of this work:
>
> 1) **Feature Alignment with Joint Multi-View Geometric Reconstruction.**
> As existing foundation models (VLMs) suffer from severe 2D feature inconsistency issues, simply combining multiple foundation models often leads to degraded accuracy rather than improvement (Lines 254-279). We test several enhanced baselines by assembling different existing modules, but found that such integration often leads to degraded performance due to conflicting features (Table 1 and Table I in the Appendix). Our approach mitigates these conflicts by establishing an implicit feature alignment at the 3D level through joint online multi-view reconstruction. This not only significantly improves the accuracy and coherence of the 3D reconstruction but also enhances overall efficiency by optimizing the 2D-to-3D lifting process.
>
>     The table below highlights this contribution, demonstrating the crucial role of our feature alignment mechanism in place of simple module combinations. "w/o features" represents a direct composition of modules via input–output connections. "w/o feature online 3D alignment" lacks our implicit 3D alignment. EA3D significantly outperforms these baselines in both performance and efficiency, yielding more consistent and high-quality 3D reconstructions. Please note that "w/o features" is time-consuming because it needs to decode all features into 2D semantic maps, and "w/o feature online 3D alignment" relies on the offline feature-GS reconstruction, where the high-dimensional features lead to slow optimization.
>
> | Method|3D Reconstruction $\uparrow$ |          3D Segmentation $\uparrow$ | Speed(fps) $\uparrow$     |
> | :--------     |    :--------:   |  :--------: |      :--------: |
> | w/o features        |    19.8   |          33.5 |          0.04 |
> | w/o feature online 3D alignment       |   23.9   |          41.9 |          0.09 |
> | Ours-full       |    **25.8**   |          **46.3** |          **0.23** |
>
> We further enhance prior and concurrent methods and modules to serve as stronger baselines, as detailed in Section C of the Appendix. The results highlight the value of our end-to-end unified feature alignment framework, grounded in online joint geometric reconstruction and optimization.
>
>   2) **Online 3D Object-aware Joint Reconstruction and Understanding.**
>   EA3D provides a novel, practical, and efficient solution by jointly performing 3D geometric reconstruction and scene understanding in a single, online pass. Existing modular systems are typically limited to either offline 3D reconstruction (Lines 31-33) or 2D image analysis (Lines 26-30), lacking the capacity for a comprehensive, online 3D scene interpretation. Moreover, concurrent 3D scene understanding and reconstruction methods typically depend on expensive geometric (Lines 81-84) and pose pre-annotation modules (Lines 195-198), which incur substantial computational overhead. In contrast, we introduce a novel online feature Gaussians framework, integrated with joint online visual odometry, which efficiently leverages past observations to reconstruct and interpret 3D scenes. Our approach delivers high-quality, efficient, and object-aware online reconstruction and understanding, while supporting a broad range of downstream tasks.
>
> ***
>
> ### [Q1. Using better pose estimation]
> Thank you for the insightful question. Yes, EA3D supports both pose-free and pose-provided inputs, where camera poses are continuously optimized during the online update. While EA3D performs well without pose pre-estimation, more accurate initial poses naturally lead to better results. Here, we replace COLMAP with an improved offline pose estimation model, GLOMAP$^{[1]}$, which further enhances the performance of EA3D.
>
> | Method|3D Reconstruction (PSNR) $\uparrow$ |3D Segmentation (mIoU) $\uparrow$ |
> | :--------|:--------:| :--------:|
> | EA3D* (w/ COLMAP)        |    25.5  |  45.9 |
> | EA3D* (w/ GLOMAP)        |    25.7  |  46.0 |
>
> >[1] Global structure-from-motion revisited.
> ***
> ### [Q2. Feature calculation and updating]
> We extract the current frame features using 2D vision foundation models (Lines 129-138) based on a series of visual encoders. The extracted current frame features are first aggregated based on their similarity to the historical online semantic cache. The current feature map is then embedded into representations through a fast feedforward step. Subsequently, new Gaussian features are incrementally initialized and added per frame (Lines 205-207), while existing ones are updated through the Online Gaussian Updating strategy (Lines 201-205), wherein both the semantic features and geometry attributes of the Gaussians are jointly refined (Lines 192-211) during the online reconstruction. We also dynamically remove high-opacity ellipsoids and apply a one-step splitting strategy to adaptively regulate Gaussian growth based on feature gradients. We will further clarify the details in our revised paper.
> ***
> ### [Q3. Improved reconstruction quality]
> The improved reconstruction performance benefits from several novel designs, including guidance from knowledge features, the dynamic optimization capability of online Feature Gaussians, and recurrent joint optimization. Specifically, the design of the online feature Gaussian and visual odometry updating (Lines 179-211) ensures maximal alignment between 3D geometry and semantics during online reconstruction, while dynamically refining under-reconstructed and over-reconstructed regions. The embedding of feature maps (Lines 160-164) guided by foundational knowledge facilitates the Gaussian representation with particular emphasis on regions of interest, enabling more detailed and accurate reconstruction of object geometry. The adaptive semantic-awareness regularization (Lines 217-224) guides the Gaussians to adaptively adjust their scale, effectively preventing oversized ellipsoids from causing visual distortions and artifacts. Meanwhile, the joint semantic-geometry optimization (Lines 225–232) introduces knowledge-guided feature and depth losses into Gaussian optimization, which helps achieve more accurate geometry and mitigates artifacts caused by overfitting. The table below demonstrates that the strategy of EA3D significantly enhances the performance of baseline reconstruction methods:
>
> | Method        |    3D Reconstruction (PSNR) $\uparrow$   | 3D Segmentation (mIoU) $\uparrow$ |
> | :--------    |    :--------:   |  :--------: |
> | Base Gaussians       |    19.4  |  32.7 |
> | w/o online optimization        |    24.6 |  44.5 |
> | w/o joint optimization        |    24.8  |  45.7 |
> | w/o semantic-aware regularization        |    25.1  |  44.3 |
> | Ours-full       |    **25.8**  |  **46.3** |
> ***
> ### [Q4. Reconstruction metrics]
> Thank you for the thoughtful reminder. The PSNR and SSIM refer to the metrics of the NVS (novel view synthesis) reconstruction. The meaning of the metrics will be further clarified in our revised paper.
> ***
> ### [Q5. Model insight]
> It can be understood from this angle, and we agree that EA3D has the capability to simultaneously reconstruct the scene online while progressively updating features. We would like to further emphasize that EA3D offers a novel perspective for aligning and aggregating 3D semantic and geometric features through online multi-view reconstruction and dynamic update strategies. Notably, EA3D is not simply a combination of separate reconstruction and perception modules, but a unified online 3D feature aggregation framework grounded in multi-view reconstruction constraints, enabling more accurate and efficient 3D scene understanding and reconstruction.
> ***
> ### [Q6. Excellent performance in downstream tasks]
> The excellent performance of EA3D primarily stems from its efficient feature fusion in 3D and the joint semantic-geometry 3D online optimization strategy. In fact, we employ the same VLM and VFM features derived from CLIP and Grounded-SAM as our baseline methods (FeatureGS, Feat-3dgs, SAM3D, OpenGaussian), as demonstrated in Table 1 and Table I in Supp. The results indicate that even when employing the same VFM models and corresponding foundational visual features, our method still achieves enhanced performance and greater stability. Our proposed efficient feature fusion and alignment, combined with the online geometric refinement strategy, effectively improve the model’s performance across multiple downstream tasks.
> ***
> If you have any further questions/concerns, please do not hesitate to let us know.
>
> Thank you very much,
>
> Authors

---

> > ### Author Response · Authors · 2025-08-02
> >
> > Dear Reviewer,
> >
> > We appreciate the time and effort you've dedicated to reviewing our submission. We have carefully addressed your comments in our response and would be glad to provide any further clarification before the discussion period ends (August 6, 11:59 pm AoE). We hope that you can consider raising the score after we address all the concerns.
> >
> > Thank you again for your time, and look forward to your replies.
> >
> > Sincerely, authors of submission 5333

---

> > > ### Comment · Reviewer_ouYe · 2025-08-06
> > >
> > > Thank you for your detailed rebuttal, which has addressed some of my concerns. I have decided to maintain my score.

---

> ### Author Response · Authors · 2025-08-09
> **Response to Reviewer ouYe**
>
> Dear Reviewer ouYe,
>
> We are pleased that our rebuttal has addressed some of your concerns. If there are any remaining issues, we would be happy to provide additional clarification or make additional revisions, in the hope that this might encourage you to raise the score. Thank you again for your valuable and constructive comments.
>
> Best regards.

---

### Note · Authors · 2025-08-15

We sincerely thank the ACs and reviewers for their thoughtful and constructive feedback.

***

In the original submission, the reviewers recognized key strengths of our work. We further clarify our core contributions as follows:

**Novelty**: The proposed unified framework for online open-world 3D object extraction presents a novel and valuable contribution to joint scene reconstruction and holistic understanding with state-of-the-art performance.

**Importance**: Our method tackles the critical challenges of scene understanding, introducing novel online, open-world, and dynamically 3D feature learning paradigms that provide valuable insights for the community.

**Effectiveness**: Extensive experiments demonstrate that EA3D delivers both high-quality and efficient geometric reconstruction and scene understanding, thereby enabling a broad range of downstream tasks.

The main points raised for further evaluation were:

**Core contributions**: We further clarified our core contributions and provided additional experimental results demonstrating that our method offers a unified and efficient framework that overcomes the limitations of existing approaches and their combinations, achieving state-of-the-art performance.

**Comparison with more baselines**: We have provided additional comparative experiments and will incorporate further discussions on their relevance.

**Model Efficiency and cost**: EA3D achieves higher quality and robustness while requiring fewer parameters and incurring lower storage overhead, as demonstrated in the rebuttal.

**Robustness of the foundation models**: Within the given time constraints, we evaluated the robustness of the employed foundation models under challenging scenarios. The results demonstrate that our method effectively mitigates the ambiguities and inconsistencies inherent in the foundation models, thereby enhancing their applicability in real-world settings.

We sincerely appreciate the reviewers' thoughtful feedback and the ACs' valuable guidance, and we are grateful for the recognition of our work and rebuttal. We believe that EA3D is well-founded both technically and practically, and contributes to advancing joint online 3D reconstruction and holistic scene understanding.

---

### Decision · Program_Chairs · 2025-09-17

**Decision:**

Accept (poster)

**Comment:**

This submission received 3 BA ratings after the reviewers read the rebuttals.   The consensus among the reviewers is that the paper presents an interesting pipeline with very good results.   It is mostly a "system-level" paper as the proposed method puts together "big blocks" (VLM, CLIP, Dino, SAM, online odometry), however, it is true, as mentioned by the authors in the rebuttals, that their geometric and semantic feature alignment is essential to the method. The authors have to update the paper as promised in the rebuttal.  Emphasising the importance of this alignment in the introduction and Figure 2 will also help the paper.